# Factors Associated with Poor Health-Related Quality of Life in Physically Active Older People

**DOI:** 10.3390/ijerph192113799

**Published:** 2022-10-24

**Authors:** Pablo Valdés-Badilla, Miguel Alarcón-Rivera, Jordan Hernandez-Martinez, Tomás Herrera-Valenzuela, Braulio Henrique Magnani Branco, Cristian Núñez-Espinosa, Eduardo Guzmán-Muñoz

**Affiliations:** 1Department of Physical Activity Sciences, Faculty of Education Sciences, Universidad Católica del Maule, Talca 3460000, Chile; 2School of Education, Universidad Viña del Mar, Viña del Mar 2520000, Chile; 3Centro de Formación Técnica Santo Tomás, Talca 3460000, Chile; 4Programa de Investigación en Deporte, Sociedad y Buen Vivir, Universidad de los Lagos, Osorno 5290000, Chile; 5Department of Physical Activity Sciences, Universidad de Los Lagos, Osorno 5290000, Chile; 6Department of Physical Activity, Sports and Health Sciences, Faculty of Medical Sciences, Universidad de Santiago de Chile (USACH), Santiago 8370003, Chile; 7Graduate Program in Health Promotion, Cesumar University (UniCesumar), Maringá 87050-900, Brazil; 8School of Medicine, University of Magallanes, Punta Arenas 6200000, Chile; 9Centro Asistencial Docente y de Investigación, Universidad de Magallanes, Punta Arenas 6200000, Chile; 10Interuniversity Center for Healthy Aging, Punta Arenas 6200000, Chile; 11School of Kinesiology, Faculty of Health, Universidad Santo Tomás, Talca 3460000, Chile; 12School of Kinesiology, Faculty of Health Sciences, Universidad Autónoma de Chile, Talca 3460000, Chile

**Keywords:** exercise, physical fitness, older adults, aging

## Abstract

This study aimed to associate morphological variables and physical fitness with health-related quality of life (HRQoL) in physically active older people. A cross-sectional study was carried out that evaluated 470 older people (89.57% female) with a mean age of 70.13 ± 6.57 years, residing in two regions of Chile. Morphological variables (body weight, bipedal height, waist circumference, body mass index, and waist to height ratio), physical fitness through the Senior Fitness Test protocol, and HRQoL using the SF-36 questionnaire were obtained. Logistic regression analysis was used to identify risk factors between morphological variables and physical fitness associated with HRQoL. The main results indicated that overweight (OR = 1.52; *p* = 0.034), a waist circumference with risk (OR = 1.56; *p* = 0.021), poor performance in the back scratch tests (OR = 1.02; *p* = 0.008) and timed up-and-go (OR = 1.19; *p* = 0.040) increased the probability of having a low general HRQoL. Also, the low performance of chair stand and arm curl tests was associated with poor physical and social dimensions of HRQoL (*p* < 0.05). In conclusion, a low HRQoL in physically active older people is associated with both morphological and physical fitness factors.

## 1. Introduction

Regular physical activity practice produces various benefits in older people [1,2], among others, a significant reduction of body weight, fat mass, waist circumference (WC), body mass index (BMI), waist to height ratio (WHR) [3], and lower fall risk [4,5]. In addition, being physically active contributes to a significant increase in muscle mass and strength, flexibility, cardiorespiratory fitness, agility, dynamic balance [3,6,7], improved perceived quality of life [6], and better cognitive function [8]. Together, these elements allow people to experience healthier aging trajectories [2,8]. On the other hand, physically active people reduce the risk of mortality from all causes by 14% and increase their life expectancy by three years compared to physically inactive people [9].

In Chile, only 29.1% of people between 60 and 69 years of age are physically active, which drops to 25.3% for those over 70 years of age [10]. Facts that added to the 94% of a sedentary lifestyle and 76.8% of overweight presented by those over 65 years of age [11] reverse a worrying scenario due to the associations that exist between sedentary behavior with lower self-reported health and lower ability to carry out activities of daily living independently [12], and among the overweight with the highest mortality (4 million annually) worldwide [13]. Therefore, compliance with the international recommendations for physical activity that propose performing between 150 to 300 min of moderate physical activity or 75 to 150 min of vigorous physical activity per week [14] implies a better state of health in older people [2,6] and possible protective effects of becoming seriously ill from COVID-19 [15]. 

On the other hand, the health-related quality of life (HRQoL) corresponds to the perception of well-being of the person regarding their physical, psychological, and social health [16]. There are several validated instruments to assess HRQoL; among them, one of the most used is the SF-36 (Short Form Health Survey) questionnaire that gives the perception of eight dimensions of health [16]. This instrument has been used in times of the COVID-19 pandemic in the Chilean population, reporting a significant association between physical inactivity and a decreased perception of general health, with an odds ratio (OR = 2.76; *p* < 0.01) [17]. Although the HRQoL is influenced by various factors [17], there is still no conclusive information regarding the most significant variables in reducing its perception in poorly studied population groups such as physically active older people. In this sense, the present study aimed to associate morphological variables and physical fitness with HRQoL in physically active older people. Based on previous studies [18,19,20] we hypothesized that higher WC, BMI, and WHR values, and lower performance on physical fitness in physically active older people are significantly associated with lower HRQoL, which would reinforce the importance of regular physical activity practice during aging.

## 2. Materials and Methods

The cross-sectional study evaluated 470 older people (89.57% female) between 60 and 89 years old (women: 69.77 ± 6.62 years; men: 73.16 ± 5.31 years), selected under an intentional non-probabilistic criterion, residing in two regions of Chile: Araucanía (n = 242) and Maule (n = 228). The sample size calculation was made from a population of 686 older people enrolled in a national physical activity program in both regions (53.2% Araucanía, 46.8% Maule). The calculation estimate was of 247 participants, where 131 corresponded to Araucanía and 116 to Maule. For this calculation, a confidence level of 95% and a margin of error of 5% were used. These analyses were performed using the GPower software (version 3.1.9.6, Franz Faul, Universiät Kiel, Germany). The inclusion criteria used were: (i) people over 60 years and under 90 years of age; (ii) enrolled in the physical activity government workshops for older people; (iii) regularly practice moderate physical activity (between 150 to 300 min a week) or vigorous physical activity (between 75 to 150 min a week) for a period greater than six months [14,21]; (iv) present the ability to understand and follow instructions in context through simple commands; and (v) functionally independent, that is, have a score equal to or greater than 43 points in the Preventive Medicine Exam for the Older People (in Spanish, EMPAM) of the Ministry of Health of Chile [22]. The older people who presented: (i) musculoskeletal disorders such as acute or chronic injuries that prevented their normal physical performance were excluded; (ii) sequela of an encephalic vascular accident; (iii) permanent or temporary contraindications to performing physical activity, (iv) presented an uncontrolled metabolic or pulmonary pathology; (v) had diagnosed coagulation problems; (vi) hemodynamic instability on the day of the assessments and; (vii) presence of diagnosed major neurocognitive disorders. 

All participants were informed of the scope of the study and signed an informed consent authorizing the use of information for scientific purposes. The study protocol was reviewed and approved by the Scientific Ethics Committee of the Universidad Autónoma de Chile (approval number: N°06-16) and developed following the Declaration of Helsinki.

### 2.1. Morphological Variables

Body weight was obtained by wearing minimal clothing and using a mechanical scale (Scale-tronix, Chicago, IL, USA; accuracy to 0.1 kg), bipedal height was measured with a stadiometer (Seca model 220, SECA, Hamburg, Germany; accuracy to 0.1 cm), and WC with a tape measure (Sanny, Brazil; accuracy to 0.1 cm). Subsequently, the BMI was calculated by dividing the body weight by the squared bipedal height (kg/m^2^), and the participants were classified according to their nutritional status as normal weight (<27.9 kg/m^2^) and overweight (28 or more kg/m^2^), following to the recommendations of the Pan American Health Organization [23] and the Ministry of Health of Chile [22]. For its part, the WHR was obtained by dividing the WC by the bipedal height [24]. Older people who had a WC value of ≥88 cm (female) and ≥102 cm (male) were considered as “at risk” [22,25] and ≥0.5 for the WHR [24]. All measurements were made following the recommendations of the International Society for Advances in Kinanthropometry (ISAK) [26]. 

### 2.2. Physical Fitness

Fitness was measured using the Senior Fitness Test protocol, previously described and validated for people between 60 to 94 years of age, functionally independent, and without health problems [27]. The order of application of the tests contemplated in the protocol was: (i) chair stand test to assess lower body strength, counting the number of repetitions in 30 s; (ii) arm curl test to assess upper body strength on the dominant limb, using a 3-lb (female) and 5-lb (male) dumbbell, counting the number of repetitions in 30 s; (iii) two-minute step test to assess cardiorespiratory fitness, recording the number of knees raises that reach at least a 70° angle on the thigh-femoral joint of each participant; (iv) chair sit-and-reach test to assess lower body flexibility, measured in cm; (v) back scratch test to assess upper body flexibility, measured in cm; and (vi) timed up-and-go test to assess agility and dynamic balance, circling a cone at 2.44 m and recording the time in seconds. The values obtained in each of the tests of the Senior Fitness Test were classified as: “below normal”, “normal”, and “above normal”, according to the normative tables for the age and sex of the participants [27].

### 2.3. Health-Related Quality of Life (HRQoL)

To assess HRQoL, the SF-36 version 2 questionnaire was used [16]. The SF-36 is a self-report instrument containing 36 questions or items that measure the attributes of eight dimensions of people’s health: physical function, physical role, body pain, vitality, social function, emotional role, mental health, and general health. Each dimension is made up of a series of questions that together give a score scale that goes from 0 (the worst state of health for that dimension) to 100 (the best state of health) [16]. To categorize these values, the arithmetic mean of the participants in each of the eight dimensions of the HRQoL assessed was calculated. From this measure, the data were dichotomized into “below the mean” and “above the mean”. This categorization procedure has been used in previous HRQoL studies [17,28].

### 2.4. Procedure

The assessments were obtained in three sessions with a recovery of 48 h between them: (i) in the first session, the morphological variables, the HRQoL, and the chair stand and arm curl tests were measured through a single repetition; (ii) in the second session, the two-minute step test (one repetition) was assessed; and (iii) in the third session the chair sit-and-reach, back scratch and timed up-and-go tests were measured, each test was repeated twice, and the best performance of each participant was reported. All measurements were carried out in the respective health centers or beneficiary sports organizations with physical activity government workshops aimed at older people through a level II anthropometrist (technical error of measurement to 0.7%) of the ISAK and researchers trained in the protocol of the Senior Fitness Test and the SF-36 questionnaire. The characteristics of the sample can be seen in Table 1.

### 2.5. Statistical Analysis

Data were analyzed with SPSS 25.0 statistical software (SPSS 25.0 for Windows, SPSS Inc., Chicago, IL, USA). Values were reported as mean ± standard deviation. The Kolmogorov–Smirnov test was used to determine the normality of the data, while the Levene’s test was used to determine the homogeneity of variance. Normal distribution was observed for all data. Associated risk factors (morphological variables and physical fitness) with low HRQoL were identified by logistic regression analysis for each of the eight dimensions assessed in the SF-36 questionnaire. These analyses were adjusted for sex, age, and nutritional status and presented as odds ratios (OR) with their respective 95% confidence intervals (95% CI) to present the magnitude of the association. The level of significance was defined as *p* < 0.05.

## 3. Results

The logistic regression analyses for the physical function, physical role, body pain, and vitality dimensions of HRQoL are summarized in Table 2. Being over 70 years of age was associated with a greater probability of presenting a decrease in HRQoL in the physical function dimensions (OR = 1.05, *p* = 0.001) and vitality (OR = 1.03, *p* = 0.022). In addition, presenting overweight (OR = 1.93; *p* = 0.002), WC with risk (OR = 1.47; *p* = 0.046), and poor performance in the chair stand (OR = 1.11; *p* = 0.001), arm curl (OR = 1.07, *p* = 0.003) and back scratch tests (OR = 1.08, *p* = 0.001) were factors associated with a decrease in HRQoL in physical fitness. On the other hand, poor performance in timed up-and-go test was associated with a decrease in HRQoL in the physical function dimensions (OR = 1.81; *p* = 0.001), physical role (OR = 1.05; *p* = 0.017) and body pain (OR = 1.29, *p* = 0.001).

Table 3 summarizes the logistic regression analysis for the social function, emotional role, mental health, and general health dimensions of the HRQoL. Being female (OR = 2.17; *p* = 0.014), presenting overweight (OR = 1.76; *p* = 0.005) and WC with risk (OR = 1.66; *p* = 0.008) was associated with greater probability of presenting low HRQoL in the social function dimension. On the other hand, being overweight (OR = 1.52; *p* = 0.034) and WC with risk (OR = 1.56; *p* = 0.021) was associated with low HRQoL in the general health dimension. Poor performance on the back scratch test was associated with decreased HRQoL in the emotional role dimensions (OR = 1.25, *p* = 0.008), mental health (OR = 1.02, *p* = 0.006) and general health (OR = 1.02, *p* = 0.008). Finally, poor performance on the timed up-and-go test was associated with a higher probability of having a low HRQoL in the social function dimensions (OR = 1.31; *p* = 0.002), emotional role (OR = 1.25; *p* = 0.005), health mental (OR = 1.35, *p* = 0.009) and general health (OR = 1.19, *p* = 0.040).

## 4. Discussion

The study’s main results indicate that some morphological variables and physical fitness factors increase the probability of having a low HRQoL in the different dimensions assessed by the SF-36 questionnaire in physically active older people. Specifically, overweight and WC with the risk increased the probability of presenting a low HRQoL in the dimensions of physical function, social function, and general health. In contrast, low performance in the chair stand and arm curl tests showed a higher probability of low HRQoL in the physical function dimension. Likewise, poor performance on the back scratch test was associated with a higher probability of low HRQoL in physical, emotional, mental, and general health dimensions. Finally, a lower-than-normal performance in the timed up-and-go test was associated with a higher probability of observing a decreased HRQoL in all measured dimensions except vitality.

Regarding morphological variables, it has previously been reported that overweight and WC above normal have been related to lower HRQoL scores (*p* < 0.001) [18]. Also, previous studies in older people have shown an association between physical fitness and HRQoL in both physical (*p* < 0.001) and mental (*p* < 0.001) aspects [19,29], being the chair stand and timed up-and-go tests the ones that are more associated with the HRQoL [19]. However, according to our knowledge, this would be one of the first studies that determine the probability of having a decreased HRQoL based on morphological variables and physical fitness in older people.

A finding observed in this study revealed that older people with higher body weight and WC were more likely to have a low HRQoL. It has been described that HRQoL corresponds to a subjective and dynamic multidimensional concept that is influenced by different factors that occur throughout life [18,30]. Among these factors, habits and lifestyles stand out as determinants of HRQoL in older people [31], which have been frequently related to morphological measures harmful to health, such as overweight and excessive accumulation of abdominal fat mass [17]. It has been suggested that unhealthy habits and lifestyles negatively affect physical, physiological, and functional aspects, causing a decrease in energy and vitality [32], which could explain the lower self-perceived HRQoL by older people with overweight and a WC above normal. Also, it has been pointed out that unhealthy habits could generate stress and anxiety, affecting the perception of HRQoL [33]. Therefore, according to our results, morphological changes caused by unhealthy habits and lifestyles could be considered risk factors for low HRQoL in the physical, social, and general health components.

Another interesting study result indicates that low physical fitness increases the probability of having a decreased HRQoL, with the agility and dynamic balance test being the most influential in HRQoL. The timed up-and-go test has been widely recognized as a measure of functional independence in older people [34], where values below normal have been related to a decline in general health (*p* < 0.05) [35], inability to perform activities of daily living (*p* < 0.05) [36], increased fall risk (OR = 42.3) [37], frailty (*p* < 0.05) [38] and decreased muscle strength (r = −0.392; *p* < 0.05) [34]. This is why the timed up-and-go test has been considered the most representative of physical functions in older people [39] and would explain our results, where older people with a lower performance in this test showed a higher risk of having a low HRQoL in seven of the eight dimensions evaluated by the SF-36 questionnaire.

In addition, it was possible to report that those over 70 years of age are more likely to have a decreased HRQoL in the physical and vitality dimensions. At the same time, the female sex was classified as a risk factor in the social function dimension of the HRQoL. This is related to previous studies showing that HRQoL is perceived more negatively at older ages and that being female has been identified as a risk factor for decreased HRQoL [17]. It has been proposed that older females face adverse and traumatic life events with negative thoughts and intrusive memories more frequently than young people and males, which could influence the perception of HRQoL [40].

Among the possible limitations of the study are: (i) not including assessments of neurophysiological mechanisms to determine the activation of the flexor-extensor muscles involved in the measurement of physical fitness; (ii) the selection of the sample (intentional non-probabilistic) that only allows analysis of association; (iii) the low number of male participants; (iv) the geographical context of the older people, which does not allow the results to be extrapolated to other realities. Among the strengths of the study, we could mention: (i) the use of the SF-36 questionnaire, which has been validated in the Chilean context to assess older people and provides situated and reliable information for health professionals; (ii) the simplicity of the morphological and physical fitness assessments, which would allow their use and implementation in physical activity programs focused on older people; (iii) the low costs to carry out these type of activities, which could be replicated in large population groups such as those found in community health centers or neighborhood associations. Given this, public policies could be organized to promote the general health of older people, including regular physical activity programs, nutrition, and psychoeducational lessons in groups, and to develop infrastructure to improve urban mobility. Emerging countries must organize public policies to develop older people’s care and stimulate self-care for all age groups; these countries (emerging countries) are experiencing a change in the age pyramid; thus, many older people will need assistance in medical care in a few years.

## 5. Conclusions

A low HRQoL in physically active older people is associated with both morphological and physical fitness factors, where overweight, WC with risk, decreased upper and lower limb strength, low upper limb flexibility. Poor agility and dynamic balance performance correspond to risk factors. From a public health perspective, it is essential to consider the factors that impact the HRQoL of older people with the intention of generating preventive and remedial strategies that can benefit the population’s health.

## Figures and Tables

**Table 1 ijerph-19-13799-t001:** Characteristics of the sample.

		Female (n = 421)			Male (n = 49)		
		Min	Mean (SD)	Max	Min	Mean (SD)	Max
Morphological variables	Body weight (kg)	40.10	69.47 (12.30)	126.20	62.10	79.07 (10.10)	100.40
	Bipedal height (m)	1.35	1.51 (0.10)	1.81	1.46	1.64 (0.06)	1.77
	WC (cm)	31.60	92.64 (12.42)	130.70	83.30	99.45 (8.86)	120.20
	BMI (kg/m^2^)	18.11	30.51 (4.93)	55.35	22.54	29.55 (3.62)	38.16
	WHR	0.22	0.61 (0.08)	0.87	0.52	0.61 (0.06)	0.73
Physical fitness	Chair stand (Rep)	4.0	16.31 (3.79)	32.0	10.0	17.71 (3.60)	25.0
	Arm curl (Rep)	5.0	22.51 (4.87)	42.0	15.0	23.33 (4.35)	37.0
	Two-minute step (Rep)	10.0	99.92 (25.15)	191.0	80.0	106.10 (13.53)	150.0
	Chair sit-and-reach (cm)	−25.4	3.88 (7.60)	34.0	−25.0	3.23 (9.10)	29.4
	Back scratch (cm)	−31.2	−8.40 (10.68)	41.9	−29.2	−12.26 (10.02)	10.0
	Timed up-and-go (s)	3.25	5.45 (1.15)	11.57	4.00	4.66 (0.59)	6.00
HRQoL	PF (score)	35	84.11 (14.10)	100	45	84.18 (11.96)	100
	RP (score)	0	91.51 (23.10)	100	0	89.29 (27.00)	100
	BP (score)	0	41.21 (10.14)	74	31	43.96 (9.28)	62
	VT (score)	15	63.63 (15.76)	90	35	64.39 (12.53)	90
	SF (score)	0	77.82 (21.21)	100	25	81.38 (21.06)	100
	RE (score)	0	93.67 (21.50)	100	0	94.56 (20.80)	100
	MH (score)	16	67.33 (14.88)	88	28	70.29 (15.17)	88
	GH (score)	8	40.64 (10.14)	80	10	40.29 (8.72)	54

SD: standard deviation. Min: minimum. Max: maximum. WC: waist circumference. BMI: body mass index. WHR: waist to height ratio. Rep: repetitions. PF: physical function. RP: role physical. BP: body pain. VT: vitality. SF: social function. RE: role emotional. MH: mental health. GH: general health. HRQoL: health-related quality of life.

**Table 2 ijerph-19-13799-t002:** Factors associated with a low HRQoL in physical function, physical role, body pain, and vitality dimensions.

(n = 470)	Physical Function				Physical Role			
	OR	CI 95%		*p*	OR	CI 95%		*p*
Female sex	0.96	0.52	1.78	0.899	0.85	0.38	1.90	0.686
Over 70 years	**1.05**	**1.02**	**1.09**	**0.001 ***	0.99	0.96	1.03	0.768
Overweight	**1.93**	**1.27**	**2.95**	**0.002 ***	1.22	0.69	2.14	0.489
WC at risk	**1.47**	**1.01**	**2.19**	**0.046 ***	1.25	0.72	2.15	0.427
WHR at risk	1.62	0.63	4.20	0.318	0.79	0.26	2.40	0.677
CS below normal	**1.11**	**1.05**	**1.17**	**0.001 ***	1.05	0.98	1.13	0.165
AC below normal	**1.07**	**1.02**	**1.11**	**0.003 ***	1.06	1.00	1.12	0.063
2-mS below normal	1.01	1.00	1.02	0.088	1.01	1.00	1.02	0.107
CSR below normal	1.02	0.99	1.05	0.134	1.01	0.98	1.05	0.551
BS below normal	**1.08**	**1.05**	**1.10**	**0.001 ***	1.01	0.98	1.03	0.705
TUG below normal	**1.81**	**1.49**	**2.20**	**0.001 ***	**1.29**	**1.05**	**1.58**	**0.017 ***
	Body pain				Vitality			
	OR	CI 95%		*p*	OR	CI 95%		*p*
Female sex	1.51	0.81	2.83	0.197	1.10	0.60	2.02	0.754
Over 70 years	0.99	0.96	1.02	0.549	**1.03**	**1.01**	**1.06**	**0.022 ***
Overweight	0.87	0.59	1.28	0.474	1.16	0.78	1.72	0.455
WC at risk	1.22	0.83	1.79	0.304	1.06	0.72	1.55	0.764
WHR at risk	0.52	0.22	1.22	0.134	0.43	0.18	1.07	0.151
CS below normal	2.14	0.43	10.69	0.356	1.03	0.98	1.08	0.235
AC below normal	1.04	1.00	1.08	0.072	1.01	0.98	1.05	0.490
2-mS below normal	1.01	1.00	1.01	0.111	1.00	1.00	1.01	0.547
CSR below normal	1.00	0.98	1.02	0.978	1.01	0.99	1.04	0.274
BS below normal	1.01	0.99	1.03	0.217	1.00	0.98	1.02	0.891
TUG below normal	**1.29**	**1.05**	**1.58**	**0.001 ***	1.02	0.87	1.20	0.783

WC: waist circumference. WHR: waist to height ratio. CS: chair stand. AC: arm curl. 2-mS: two-minute step. CSR: chair sit-and-reach. BS: back scratch. TUG: timed up-and-go. HRQoL: health-related quality of life. Data presented as odds ratio (OR) and their respective 95% CI. The analysis was adjusted for sex, age, and nutritional status. An OR > 1 indicates that there is a higher probability of having a poor HRQoL. *: *p*-value < 0.05 was considered significant.

**Table 3 ijerph-19-13799-t003:** Factors associated with a low HRQoL on social function, emotional role, mental health, and general health dimensions.

(n = 470)	Social Function				Emotional Role			
	OR	CI 95%		*p*	OR	CI 95%		*p*
Female sex	**2.17**	**1.17**	**4.03**	**0.014 ***	1.43	0.49	4.17	0.509
Over 70 years	0.98	0.95	1.01	0.121	0.99	0.95	1.04	0.678
Overweight	**1.76**	**1.19**	**2.59**	**0.005 ***	1.38	0.72	2.64	0.330
WC at risk	**1.66**	**1.14**	**2.42**	**0.008 ***	1.43	0.77	2.67	0.256
WHR at risk	0.81	0.35	1.89	0.629	1.31	0.30	5.75	0.721
CS below normal	1.04	0.99	1.09	0.093	1.04	0.96	1.13	0.318
AC below normal	1.00	0.97	1.04	0.819	1.04	0.97	1.10	0.260
2-mS below normal	1.01	1.00	1.02	0.103	1.01	1.00	1.02	0.065
CSR below normal	1.01	0.99	1.04	0.358	0.99	0.96	1.03	0.688
BS below normal	1.02	1.00	1.04	0.055	**1.04**	**1.01**	**1.06**	**0.008 ***
TUG below normal	**1.31**	**1.10**	**1.56**	**0.002 ***	**1.25**	**1.00**	**1.57**	**0.005 ***
	Mental health				General health			
	OR	CI 95%		*p*	OR	CI 95%		*p*
Female sex	1.25	0.67	2.35	0.487	0.85	0.47	1.53	0.583
Over 70 years	0.98	0.95	1.01	0.156	1.02	1.00	1.05	0.106
Overweight	1.23	0.83	1.85	0.306	**1.52**	**1.03**	**2.25**	**0.034 ***
WC at risk	1.00	0.68	1.47	0.995	**1.56**	**1.07**	**2.28**	**0.021 ***
WHR at risk	0.76	0.33	1.77	0.524	2.13	0.86	5.26	0.103
CS below normal	1.02	0.97	1.07	0.562	1.01	0.96	1.06	0.831
AC below normal	1.01	0.97	1.05	0.671	1.06	1.02	1.10	0.085
2-mS below normal	1.00	0.99	1.01	0.925	1.00	1.00	1.01	0.302
CSR below normal	1.01	0.99	1.03	0.447	1.02	0.99	1.04	0.153
BS below normal	**1.02**	**1.00**	**1.04**	**0.006 ***	**1.02**	**1.01**	**1.04**	**0.008 ***
TUG below normal	**1.35**	**1.11**	**1.64**	**0.009 ***	**1.19**	**1.01**	**1.40**	**0.040 ***

WC: waist circumference. WHR: waist to height ratio. CS: chair stand. AC: arm curl. 2-mS: two-minute step. CSR: chair sit-and-reach. BS: back scratch. TUG: timed up-and-go. HRQoL: health-related quality of life. Data presented as odds ratio (OR) and their respective 95% CI. The analysis was adjusted for sex, age and nutritional status. An OR > 1 indicates a higher probability of having a poor HRQoL. *: *p*-value < 0.05 was considered significant.

## Data Availability

The datasets generated during and/or analyzed during the current research are available from the Corresponding author upon reasonable request.

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
