# Peer review of "Factors Associated with Poor Health-Related Quality of Life in Physically Active Older People"

_ijerph, 2022, doi:10.3390/ijerph192113799_

Round 1
Reviewer 1 Report
Dear Author
1. In the Introduction you mentioned "people. Base on previous studies [18,19] we hypothesized that higher WC, BMI, and WHR values, and lower performance on physical fitness in physically active older people are significantly associated with lower HRQoL.". So why to confirm that again when it is already known? What is new in your results regarding WC, BMI and WHR.
2. Line 80 - The inclusion criteria was - regularly practice moderate physical activity (between 150 to 300 minutes a week) or vigorous physical activity (between 75 to 150 minutes a week) for a period greater than six months. How do you assess this? Just subjective evidence? Does this inclusion criteria have an impact on your study results?
3. Your study was cross-sectional. You mentioned that the assessments were obtained in three sessions. What was the time gap between the 3 sessions? Was the same time gap followed for all the patients equally?
Reviewer 2 Report
This cross-sectional study was conducted with the aim to associate morphological variables and physical fitness with HRQoL in physically active older people. These are my comments and suggestions:
Abstract:
Nicle written, just one comment: change the phrase "a waist circumference with risk" with specific number.
Introduction:
Introduction is nicely written with clearly stated aim of the research. Theoretical framework is adequately addressed.
Methods:
How did you assess that participants regularly practiced moderate PA? Line 85: in the sentence regarding exclusion criteria for all conditions mentioned should be stated that this was exclusion criteria. It is stated only for the first criterion. How did you estimate your sample size? Table 1: include range (min-max) separately for male and female participants.
Results:
Values which you considered below normal are not explained in the methods. Please, specify what is, for example, TUG below normal.
Discussion:
Most of your sample was female. This is a limitation of the study, so please mention it.
